# The Impact of Polymers on Enzalutamide Solid Self-Nanoemulsifying Drug Delivery System and Improved Bioavailability

**DOI:** 10.3390/pharmaceutics16040457

**Published:** 2024-03-26

**Authors:** Su-Min Lee, Jeong-Gyun Lee, Tae-Han Yun, Chul-Ho Kim, Jung-Hyun Cho, Kyeong-Soo Kim

**Affiliations:** 1Department of Pharmaceutical Engineering, Gyeongsang National University, 33 Dongjin-ro, Jinju 52725, Republic of Korea; m8121@naver.com (S.-M.L.); leepipi87@naver.com (J.-G.L.); xogks7702@naver.com (T.-H.Y.); chkim63@gnu.ac.kr (C.-H.K.); 2Department of Pharmaceutical Engineering, Dankook University, 119 Dandae-ro, Dongnam-gu, Cheonan 31116, Republic of Korea

**Keywords:** enzalutamide, solid self-nanoemulsifying drug delivery system, Kollidon VA64, solubility, bioavailability

## Abstract

Enzalutamide (ENZ), marketed under the brand name Xtandi^®^ as a soft capsule, is an androgen receptor signaling inhibitor drug actively used in clinical settings for treating prostate cancer. However, ENZ’s low solubility and bioavailability significantly hinder the achievement of optimal therapeutic outcomes. In previous studies, a liquid self-nanoemulsifying drug delivery system (L-SNEDDS) containing ENZ was developed among various solubilization technologies. However, powder formulations that included colloidal silica rapidly formed crystal nuclei in aqueous solutions, leading to a significant decrease in dissolution. Consequently, this study evaluated the efficacy of adding a polymer as a recrystallization inhibitor to a solid SNEDDS (S-SNEDDS) to maintain the drug in a stable, amorphous state in aqueous environments. Polymers were selected based on solubility tests, and the S-SNEDDS formulation was successfully produced via spray drying. The optimized S-SNEDDS formulation demonstrated through X-ray diffraction and differential scanning calorimetry data that it significantly reduced drug crystallinity and enhanced its dissolution rate in simulated gastric and intestinal fluid conditions. In an in vivo study, the bioavailability of orally administered formulations was increased compared to the free drug. Our results highlight the effectiveness of solid-SNEDDS formulations in enhancing the bioavailability of ENZ and outline the potential translational directions for oral drug development.

## 1. Introduction

Oral administration is the most frequently used route for drug delivery. However, approximately 40% of the drugs currently on the market and approximately 70% of new drug candidates are known to be poorly water-soluble [1,2,3]. These poorly water-soluble drugs exhibit reduced oral absorption rates due to their low solubility. This leads to increased dosages, decreased patient compliance, and the need for alternative delivery routes [4,5]. Therefore, enhancing the solubility of poorly soluble drugs for oral administration represents a significant challenge [6].

Enzalutamide (ENZ) is an androgen receptor signaling inhibitor that both inhibits the proliferation and induces the death of prostate cancer cells [7,8,9]. Despite these therapeutic advantages, ENZ exhibits low water solubility (Biopharmaceutics Classification System, BCS class II) due to its lipophilic properties. Additionally, even when dissolved in an amorphous form in aqueous solutions, it tends to quickly recrystallize [10,11]. In a previous study, a self-nanoemulsifying drug delivery system (SNEDDS), which is a lipid-based drug delivery system, was applied to reflect the characteristics of ENZ and develop a liquid SNEDDS formulation (Labrafac PG, Solutol HS15, and Transcutol P at a weight ratio of 1:8:1), achieving approximately a twofold improvement in oral absorption compared to commercial products [12]. However, the liquid SNEDDS generally has stability issues because of precipitation or phase separation due to the agglomeration of particles. Furthermore, limitations, such as the limited storage conditions, complex process control, and high manufacturing costs of soft capsules, must be addressed [13,14,15,16,17]. To overcome these challenges, research on SNEDDS solidification using methods such as melt extrusion, adsorption to solid carriers, wet granulation using a high-speed mixer, spray drying, and supercritical fluid-based methods is actively ongoing [17,18,19,20,21,22]. In this study, a solid form of liquid SNEDDS (S-SNEDDS) was developed using spray drying, incorporating a solidification carrier and a polymer to restrain recrystallization. This polymer helps ENZ remain either molecularly dissolved or in an amorphous state in the S-SNEDDS directly after spray drying. Subsequently, it helps to induce and maintain a supersaturated solution, preventing rapid recrystallization of the drug and thereby enhancing drug absorption in vivo [10]. Consequently, the physicochemical properties of the S-SNEDDS formulation with added polymer were evaluated, and improvements in bioavailability were assessed through in vivo rat pharmacokinetics (PK) studies.

## 2. Materials and Methods

### 2.1. Materials

ENZ was purchased from MSN Laboratories Private Ltd. (Telangana, India). Copovidone (Abbreviated as Kollidon VA64) and colloidal silica were supplied by Boryung Pharmaceutical. Co., Ltd. (Seoul, Republic of Korea). Hydroxypropylcellulose (Abbreviated as HPC-L) was supplied by Cosmax Pharmaceutical. Co., Ltd. (Cheongju, Republic of Korea). Hydroxypropylmethylcellulose phthalate (HPMCP HP55) was provided by Hanmi Pharmaceutical. Co., Ltd. (Seoul, Republic of Korea). N,N-dimethylaminoethyl methacrylate (abbreviated as Eudragit E PO) was obtained from Evonik (Essen, Germany). Propylene glycol dicaprylocaprate (abbreviated as Labrafa PG) and Diethylene glycol monoethyl ether (abbreviated as Transcutol P) were acquired from Gattefosse (St. Priest, France). Macrogoli 15 hydroxystearas (abbreviated as Solutol HS15) was sourced from BASF (Ludwigshafen, Germany). Sodium carboxymethyl cellulose was purchased from Duksan Chemicals (Ansan, Republic of Korea). Acetonitrile, ammonium acetate, and acetone were purchased from Daejung Chemicals (Siheung, Republic of Korea). Other chemicals used were of the appropriate class for analysis.

### 2.2. HPLC Analysis Condition

ENZ concentration test was performed using HPLC (Agilent 1260 series, Agilent Technologies, Santa Clara, CA, USA) equipped with a UV-Vis detector, high-pressure gradient pump, and a VDSpher 100 C18 M-E column (150 × 4.6 mm, 5 µm). The mobile phase consisted of acetonitrile and 20 mM ammonium acetate aqueous solution (55/45, *v*/*v*), adjusted to pH 4.6 using acetic acid. The HPLC analysis was performed at a flow rate of 1.3 mL/min, injection volume of 10 μL, and a UV wavelength of 235 nm [23]. Data collection and processing were performed using OpenLab CDS CS C.01.08 ChemStation software.

### 2.3. Saturation Solubility Test

The solubility of ENZ in 1% (*w*/*v*) polymer solutions dissolved or dispersed in D.W. (distilled water) at pH 1.2 and pH 6.8 was evaluated to select the appropriate polymer for the development of S-SNEDDS. An excess amount of crystalline or amorphous ENZ powder (10 mg) was added to 1 mL of each polymer solution, followed by 30 min of vortexing and sonication. Subsequently, to achieve a supersaturated solution, the mixture was shaken for 72 h in a 37 °C water bath at 75 rpm and was frequently mixed in a shaking water bath. The samples were centrifuged at 13,500 rpm for 10 min, and then the supernatant was filtered via a 0.45 μm syringe filter to eliminate insoluble ENZ. The obtained solution was diluted twofold in the HPLC mobile phase, and the concentration of ENZ in each solution was assessed using the aforementioned HPLC system (Agilent 1260 series, Agilent Technologies, Santa Clara, CA, USA) [24].

### 2.4. Preparation of ENZ-Loaded Solid SNEDDS (S-SNEDDS)

ENZ-loaded S-SNEDDS formulations were prepared using a lab-scale mini spray-dryer (Yamato ADL311SA, Yamato Scientific Co., Ltd., Tokyo, Japan). Based on the solubility test and the S-SNEDDS manufacturing process, Eudragit E PO, HPMCP HP55, HPC-L, and Kollidon VA64 were selected as the solid carriers for solidifying liquid SNEDDS. A total of 1 g of liquid SNEDDS vehicles containing 40 mg of ENZ and each polymer was dissolved in 20 g of acetone (Table 1 and Table 2). Then, colloidal silica, serving as the solidification carrier, was uniformly suspended in these resulting solutions and spray-dried under the following conditions: inlet temperature of 80 °C, outlet temperature of 60 °C, feeding flow rate of 3 mL/min, and atomizing air pressure of 0.1 MPa.

### 2.5. Aqueous Solubility and Dissolution

#### 2.5.1. Aqueous Solubility

An excess of ENZ-loaded S-SNEDDS (equivalent to 10 mg of ENZ) was separately added to 5 mL of distilled water, mixed appropriately using vortex, and then shaken on a mechanical shaker with a water bath at 37 °C for 72 h. These samples were centrifuged at 13,500 rpm for 10 min, diluted 2-fold with acetonitrile, and filtered using a 0.2 μm syringe filter. The concentration of ENZ in the filtered sample was analyzed using the HPLC conditions described in Section 2.2.

#### 2.5.2. Dissolution

The dissolution test result of ENZ-loaded S-SNEDDS formulations was evaluated using a dissolution tester (USP apparatus II) and a dissolution tester (Dream Test, Seoul, Republic of Korea). The dissolution test medium was maintained at 37 ± 0.5 °C, and the paddle speed was kept constant at 50 rpm throughout the testing period. ENZ powder and S-SNEDDS formulations, each containing an equivalent of 40 mg of ENZ, were exposed to 900 mL of dissolution media at pH 1.2, 4.0, and 6.8, and in distilled water. Each sample was aliquoted from dissolution medium at predetermined time point (5, 10, 15, 20, 30, 45, 60, 90, and 120 min). Each aliquoted sample was filtered through a 0.2 µm syringe filter and diluted 2-fold with acetonitrile. The concentration of ENZ in the filtered sample was assayed using the HPLC conditions described in Section 2.2.

### 2.6. Solid Characterization

X-ray powder scattering measurements were performed at room temperature using an X-ray diffractometer (D/MAX-2500, Rigaku, Japan). Monochromatic Cu Kα radiation (λ = 1.54178 Å) was used at 100 mA and 40 kV. An angular increment of 0.02° per second was selected over the 2θ angle range from 2° to 60° [25].

### 2.7. Thermal Analysis

Differential scanning calorimetry (DSC) measurements were performed using a Thermal Analysis system with a DSC Q200 (TA Instruments, New Castle, DE, USA). Accurately weighed samples were positioned in standard aluminum pans, and nitrogen was used as the purge gas. All samples were scanned at a temperature ramp rate of 10 °C/min, with the heat flow measured from 20 °C to 260 °C [26].

### 2.8. Surface Morphological Features

The surface characteristics and shape of the samples were examined by using a Tescan-MIRA3 scanning electron microscope (TESCAN KOREA, Seoul, Republic of Korea). Double-sided adhesive tape was utilized to fix the samples. To render the samples electrically conductive, a platinum coating (4 min at 25 mA) was applied using an EmiTech Sputter Coater (K575X) at a speed of 6 nm/min under vacuum (7 × 10^−3^ mbar) [27].

### 2.9. Particle Size Distribution

The particle size of S-SNEDDS formulations (SNE_S1 and SNE_VA4) dispersed in 300 mL of distilled water was analyzed using a particle size analyzer (Malvern Korea, Malvern, UK). The analysis conditions were set to a wavelength of 633 nm, a scattering angle of 173 °C, and a temperature of 37 °C. The hydrodynamic diameter was analyzed in triplicate [12,28].

### 2.10. Pharmacokinetics

#### 2.10.1. Animal Care

Male Sprague–Dawley rats, age 9–10 weeks and weight 250 ± 20 g were housed in cages at room temperature (45–60% RH) and had access to drinking water ad libitum for 24 h. The Institutional Animal Care and Use Committee (IACUC) at Gyeongsang National University (approval no. GNU-230719-R0149) approved the animal study protocols, in accordance with the NIH Policy and the Animal Welfare Act.

#### 2.10.2. Oral Administration and Blood Sampling

Rats were non-selectively divided into three groups, with each group consisting of six rats. Each rat was anesthetized, and the right femoral artery was cannulated using polyethylene tubing. Rats were orally administered ENZ powder and SNE_VA4 dispersion (50 mg/mL) in 0.5% carboxymethyl cellulose at a dose of 50 mg/kg. Blood (350 μL) was collected via the cannulated tube at the following indicated time points: 0.25, 0.5, 1, 2, 3, 6, 9, and 12 h, and 1, 2, and 3 days, and immediately centrifuged at 13,500 rpm for 15 min at 4 °C. Finally, plasma was separated from the supernatant and kept at −20 °C until analysis [10,12,29].

#### 2.10.3. Blood Analysis

The collected plasma samples (150 μL) were deproteinized with acetonitrile (300 μL) in 2 mL microtubes. Afterwards, 50 μL of internal standard solution (tadalafil in acetonitrile, 20 μg/mL) was added, and the tubes were vortexed for 1 min and centrifuged at 13,500 rpm for 15 min. The concentration of ENZ in the supernatant was quantified using the HPLC analysis with an injection volume of 40 uL under the analysis conditions described in Section 2.2 [12,30]. Non-compartmental analysis was used to calculate the maximum plasma concentration of the drug (C_max_), the area under plasma concentration versus time curve from 0 to 72 h (AUC_0–72_), the time taken to reach the maximum plasma concentration (T_max_), the elimination rate constant (K_el_), and the half-life (t_1/2_). A Student’s *t*-test was performed to determine statistically significant differences (*p* < 0.05) between test groups. All data are expressed as mean ± SD.

## 3. Results and Discussions

### 3.1. Solubility of ENZ

Through prior research, an optimal liquid SNEDDS formulation was developed that improved the oral absorption of ENZ by approximately twofold [12]. However, during the powdering process, the solubility decreased due to ENZ recrystallization. An experiment was conducted to select a recrystallization inhibitor to solve the decrease in solubility that occurs during the solidification process of the liquid SNEDDS. As mentioned in the introduction, polymers can be used to maintain the amorphous state of a drug when dissolved in an aqueous solution. Specifically, ENZ is a drug that recrystallizes quickly, a tendency that becomes more pronounced when an excess amount of the drug is added to an aqueous solution. Therefore, the solubility test aimed to identify the polymer that could maintain the highest drug concentration by inhibiting recrystallization, after immersing an excess amount of the drug in various 1% (*w*/*v*) polymer solutions for 3 days. As a result, the saturation solubility of crystalline and amorphous ENZ showed no difference across various pH levels. The polymers used did not significantly increase the saturation solubility of crystalline ENZ (Figure 1). However, HPC-L, HPMC 2910, Kollidon VA64, HPMCP HP55, and Eudragit E PO demonstrated a more than twofold increase in solubility for amorphous ENZ under all pH conditions (excluding Eudragit E PO in pH 6.8). Acetone is known to be an excellent solvent for the solubility of ENZ [12], and to use acetone for the manufacturing of ENZ-loaded S-SNEDDS, HPMC, which does not dissolve in acetone, was excluded. Therefore, based on the solubility tests, HPC-L, Kollidon VA64, HPMCP HP55, and Eudragit E PO, which are soluble in acetone, were selected as polymers for further research.

### 3.2. Aqueous Solubility of Prepared ENZ-Loaded S-SNEDDS Formulations

S-SNEDDS formulations were prepared using the selected HPC-L, HPMCP HP55, Kollidon VA64, and Eudragit E PO with ENZ at a 1:1 weight ratio, and solubility tests were conducted for both initial and long-term storage over 2 months (Figure 2). Initially, SNE_S1 (213.06 ± 3.44 mg/mL), SNE_EP1 (193.63 ± 0.41 mg/mL), and SNE_HP1 (229.99 ± 12.98 mg/mL) showed no significant differences in solubility. SNE_L1 (361.22 ± 0.78 mg/mL) and SNE_VA1 (421.56 ± 0.78 mg/mL) exhibited a solubility approximately 1.7-fold and 1.9-fold higher than SNE_S1, respectively. Moreover, after 2 months of long-term storage, the solubility of the S-SNEDDS formulations containing polymer did not decrease compared to the initial measurements. However, SNE_S1, which did not contain polymer, showed a solubility decrease of approximately 29% (213.06 mg/mL -> 151.39 mg/mL) after 2 months of long-term storage.

### 3.3. Dissolution Profile of Prepared ENZ-Loaded S-SNEDDS Formulations

The in vitro dissolution test of ENZ from the S-SNEDDS formulations was performed in pH 1.2 and 6.8 solutions, simulating gastrointestinal pH conditions. Under both pH 1.2 and 6.8 conditions, the SNE_S1 formulation without polymer reached approximately 70% dissolution 15 min after the start, but the dissolution rate rapidly decreased. This decrease is attributed to the loss of solubility advantage due to the inherent tendency of the amorphous drug to crystallize in a supersaturated solution, showing 40.37 ± 3.30% and 24.43 ± 2.96% dissolution at 120 min, respectively (Figure 3). In contrast, formulations containing polymers, such as SNE_EP1, SNE_HP1, SNE_L1, and SNE_VA1, tended to maintain their dissolution rate without a sharp decrease until 120 min, with the SNE_VA1 formulation showing the highest dissolution rate. An amorphous formulation comprising a molecular-level blend of drug and polymer improves solubility compared to a crystalline system, and the tendency for the drug to crystallize in a supersaturated state is mitigated [31,32,33,34]. Based on these results, Kollidon VA64 was selected as a recrystallization inhibitor, and the drug dissolution pattern from S-SNEDDS according to the Kollidon VA64 content was confirmed. In pH 1.2 solutions, the dissolution rates of SNE_VA1, VA2, VA3, VA4, and VA5 at 15 min from the start were 69.79 ± 0.94%, 66.01 ± 0.05%, 93.86 ± 0.14%, 103.28 ± 0.60%, and 104.65 ± 1.66%, respectively. A similar trend was observed in a pH 6.8 solution, showing that the dissolution rate tended to increase as the content of Kollidon VA64 in the S-SNEDDS formulation increased (Figure 4). SNE_VA4 and SNE_VA5 showed similar dissolution rates; thus, SNE_VA4 was selected as the optimized S-SNEDDS, considering the formulation’s weight. Additionally, the dissolution patterns of crystalline ENZ powder were analyzed and compared with SNE_S1 and SNE_VA4 at pH 1.2, 4.0, and 6.8, and in D.W (Figure 5). Under all conditions, the dissolution of ENZ powder was scarcely detected owing to its low solubility. SNE_S1 exhibited a tendency for the dissolution rate to decrease from 20 to 30 minutes after dissolution compared to initial. On the other hand, the SNE_VA4 formulation exhibited complete drug release within the first 10 min of dissolution, without a significant decrease until 120 min, suggesting the potential for enhancing oral absorption.

### 3.4. Physicochemical Properties

The powder X-ray diffraction (P-XRD) patterns are shown in Figure 6A. The ENZ powder exhibited a highly crystalline structure, characterized by several distinctive peaks. However, in solidified SNEDDS, the unique pattern of ENZ disappeared, indicating that the drug was either molecularly dissolved or in an amorphous state within the SNEDDS.

The thermogram of ENZ powder, as shown in Figure 6B, reveals a significant sharp exothermic peak at 200 °C, attributed to its decomposition at the melting temperature, thus demonstrating its crystalline characteristics. In contrast, S-SNEDDS formulations (SNE_S1 and SNE_VA4) exhibited no intrinsic peaks.

FE-SEM images confirmed the surface morphology of crystalline ENZ, Kollidon VA64, and S-SNEDDS formulations (Figure 7). ENZ powder exhibited a polygonal crystalline structure, while Kollidon VA64 showed a rounded circular appearance. Unlike the ENZ powder, the ENZ-loaded S-SNEDDS formulations showed significant changes in particle shape and surface topography. The crystalline structure of ENZ disappeared in the S-SNEDDS formulations. Consequently, the improvement in solubility of the S-SNEDDS formulation was a result of applying formulation technology to transform the drug from a highly stabilized crystalline state to a high-energy amorphous state, which also suggests the possibility of improved oral absorption [35,36].

### 3.5. Particle Size Distribution

SNEDDS is a solubilization technology that can improve drug solubility by increasing the surface area through the provision of nano-sized particles. A liquid SNEDDS, optimized through previous research, was manufactured as a stable emulsion with a particle size and PDI of 15.92 ± 0.13 nm and 0.03 ± 0.01 nm, respectively. Thus, the particle size and PDI of the optimized S-SNEDDS, containing a solidification carrier and a recrystallization inhibitor, were compared with those of the liquid SNEDDS (Figure 8). As a result, the particle size and PDI values for SNE_S1 were 23.25 ± 0.58 nm and 0.23 ± 0.02, respectively, and for SNE_VA4 they were 22.92 ± 0.22 nm and 0.21 ± 0.01, respectively, showing results that were not significantly different from those of the liquid SNEDDS. Therefore, it was confirmed that the solidified SNEDDS was equivalent to the liquid SNEDDS in terms of these measures.

### 3.6. Pharmacokinetics

The PK profile of an ENZ-loaded S-SNEDDS formulation (SNE_VA4) administered orally to rats at 50 mg/kg was compared with that of rat ENZ powder. The changes in mean plasma concentration and pharmacokinetic parameters (Cmax, Tmax, AUC, Kel, t1/2) of ENZ are shown in Figure 9 and Table 3, respectively. The results indicated that the SNE_VA4 formulation led to significantly higher plasma concentrations compared to the free drug (*p* < 0.05). Specifically, the C_max_ (8.9 ± 2.0 μg/mL) and AUC (274.4 ± 47.6 h × μg/mL) values for the group treated with SNE_VA4 were 6.4-fold and 7.3-fold higher than those for the group treated with the crystalline ENZ powder, respectively. The improved oral bioavailability of ENZ in solidified SNEDDS can be explained by the formation of nano-sized emulsions during the gastrointestinal tract and the effect of maintaining the amorphous state of the drug in supersaturated solution by the polymer. The improved oral bioavailability of ENZ in solidified SNEDDS is explained by the fact that the S-SNEDDS spontaneously forms nano-emulsions upon contact with aqueous solutions in the gastrointestinal tract, and the polymer maintains an amorphous state by alleviating the tendency of the drug to crystallize in a supersaturated state.

## 4. Conclusions

In this study, an ENZ-loaded S-SNEDDS containing Kollidon VA64 as a recrystallization inhibitor was developed. The screening of various polymers revealed that the use of polymers increased the saturation solubility of amorphous ENZ; among the polymers tested, VA64 exhibited the most significant enhancement in solubility. Moreover, when the drug and VA64 were used in a ratio exceeding 1:4, there was almost no reduction in dissolution. This formulation significantly increased the solubility of ENZ by maintaining its amorphous state in a supersaturated solution with Kollidon VA64, resulting in the release of all of the drug within 10 min. Upon dispersion in an aqueous solution, it formed a stable emulsion similar to the liquid SNEDDS, with a particle size of approximately 22 nm. Furthermore, the in vivo study demonstrated a prominent increase in bioavailability by more than sixfold after oral administration of the optimal ENZ S-SNEDDS formulation compared to ENZ powder. Therefore, the potential benefits of this formulation, with improved oral absorption, make it a promising candidate for clinical use. With further investigation and regulatory approval, it holds the potential to significantly advance the treatment of prostate cancer.

## Figures and Tables

**Figure 1 pharmaceutics-16-00457-f001:**
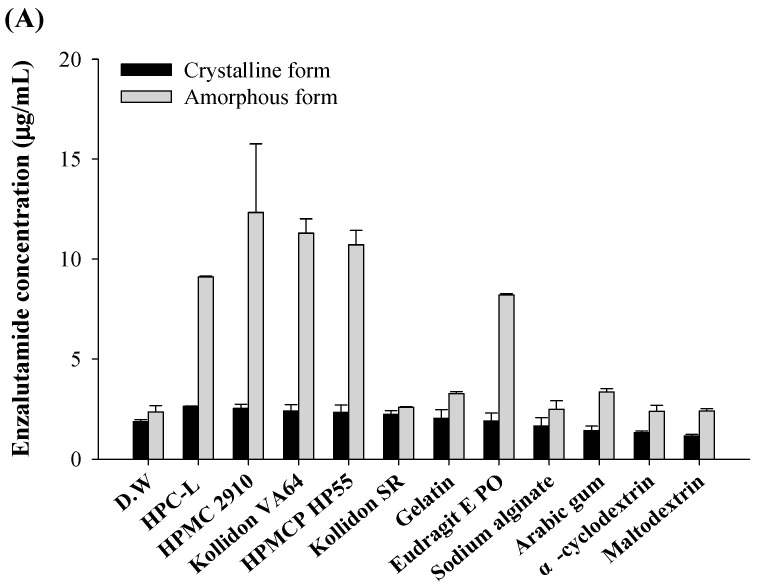
The solubility of ENZ in various 1% (*w*/*v*) polymer solutions dissolved in (**A**) D.W., (**B**) pH 1.2, and (**C**) pH 6.8.

**Figure 2 pharmaceutics-16-00457-f002:**
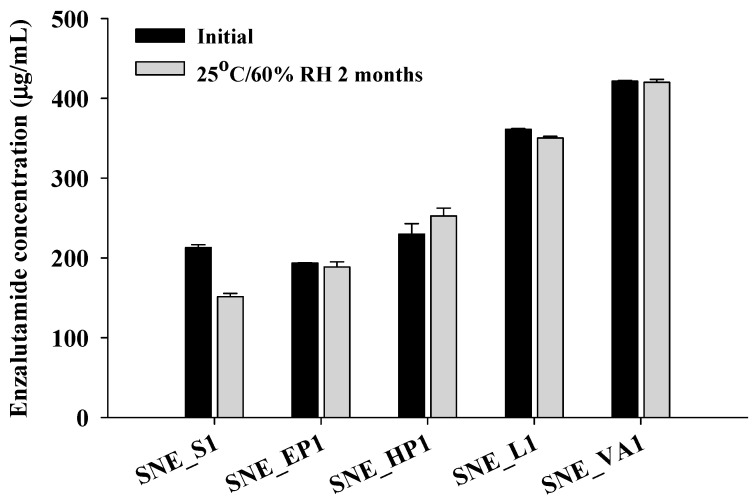
The ENZ solubility from S-SNEDDS formulation under initial and long-term storage conditions.

**Figure 3 pharmaceutics-16-00457-f003:**
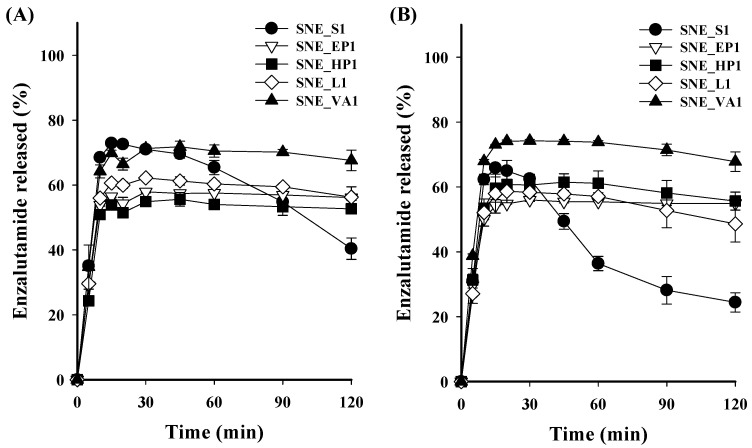
The ENZ dissolution profile from S-SNEDDS formulation for various polymers under (**A**) pH 1.2 and (**B**) pH 6.8.

**Figure 4 pharmaceutics-16-00457-f004:**
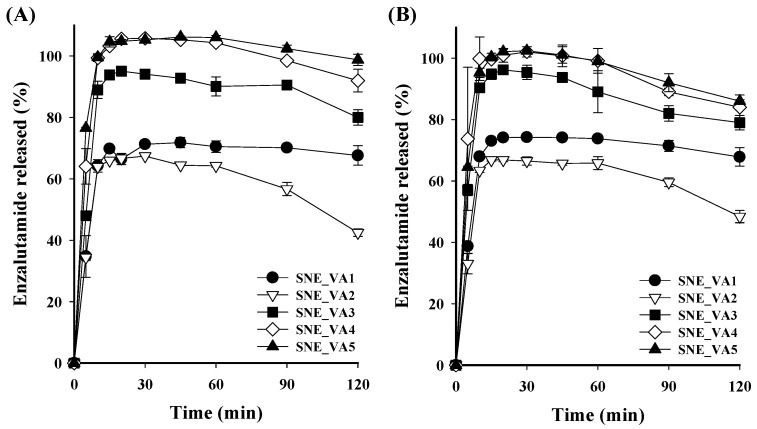
The ENZ dissolution profile from S-SNEDDS formulation for Kollidon VA64 content under (**A**) pH 1.2 and (**B**) pH 6.8.

**Figure 5 pharmaceutics-16-00457-f005:**
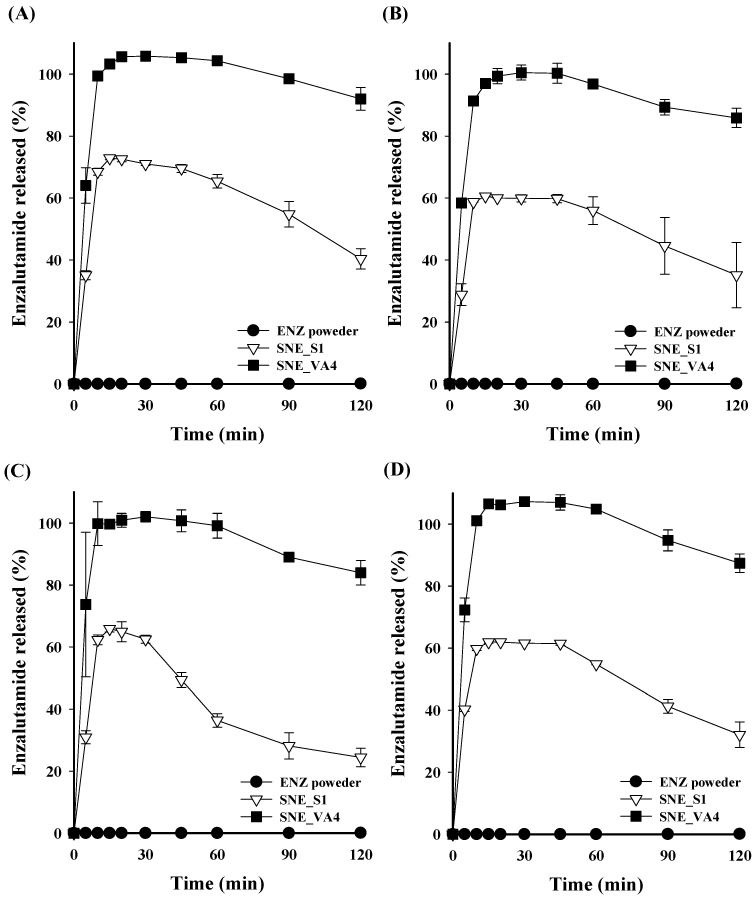
In vitro dissolution profiles of ENZ powder and SNE_VA4. Dissolution profile under (**A**) pH 1.2, (**B**) pH 4.0, (**C**) pH 6.8, and (**D**) distilled water.

**Figure 6 pharmaceutics-16-00457-f006:**
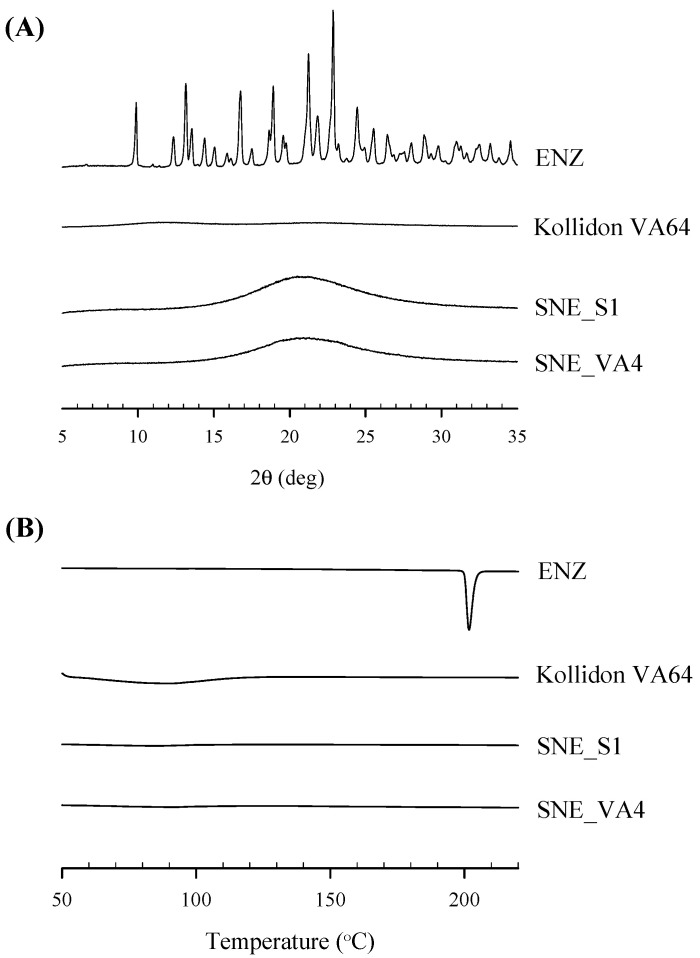
Powder X-ray diffractogram (**A**) and differential scanning calorimetric thermogram (**B**).

**Figure 7 pharmaceutics-16-00457-f007:**
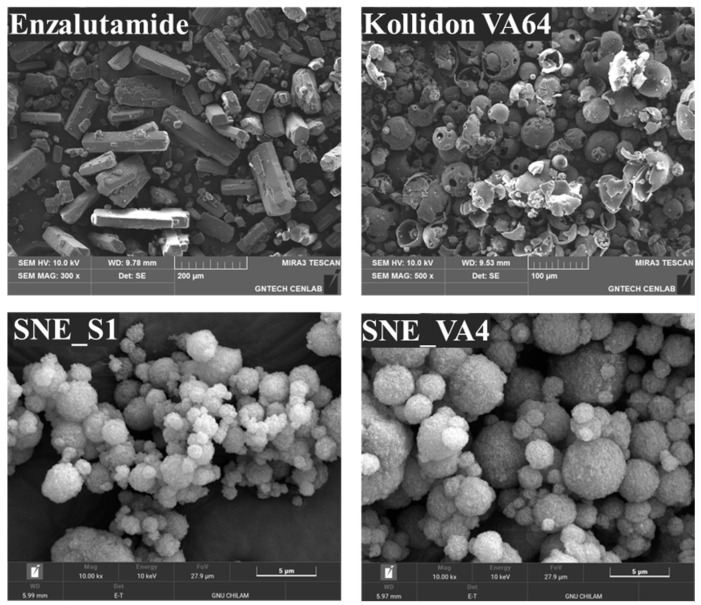
FE-SEM images of raw enzalutamide, Kollidon VA64, and enzalutamide-loaded solid-SNEDDS. (×10,000).

**Figure 8 pharmaceutics-16-00457-f008:**
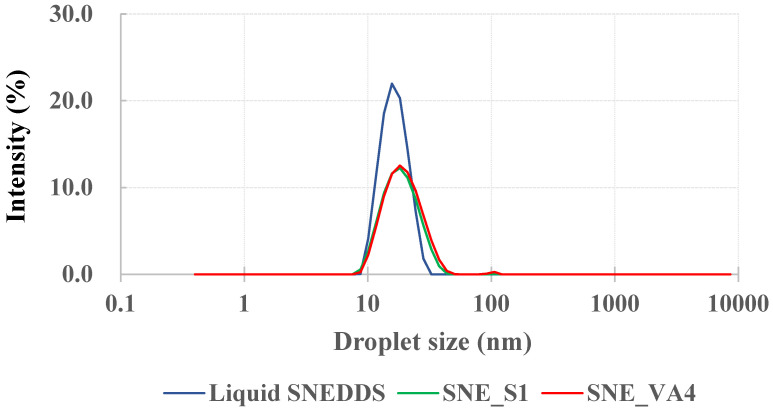
Intensity size distribution curve of L-SNEDDS and S-SNEDDS formulations.

**Figure 9 pharmaceutics-16-00457-f009:**
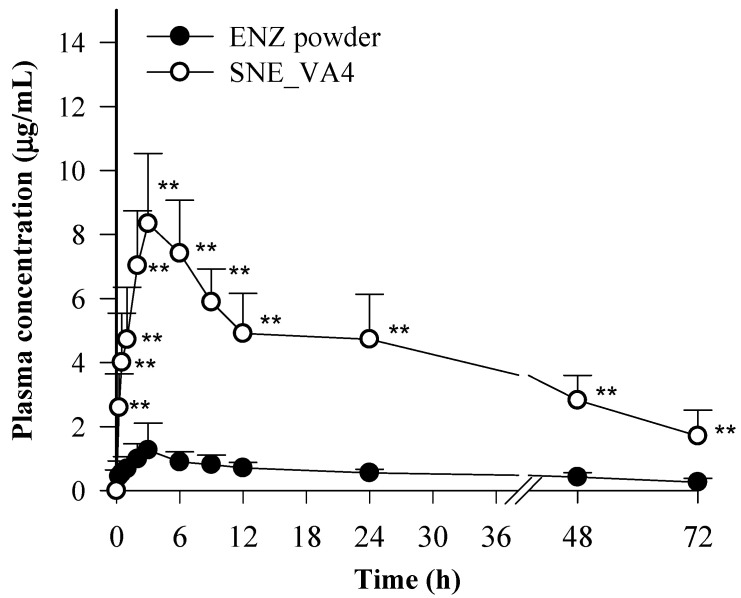
Plasma concentration–time profiles in rats after oral administration of ENZ powder, and optimized S-SNEDDS formulation (SNE_VA4); ** *p* < 0.05 compared with ENZ powder.

**Table 1 pharmaceutics-16-00457-t001:** Composition of ENZ S-SNEDDS formulations loaded with various polymers.

Formulation (mg)	SNE_S1	SNE_EP1	SNE_HP1	SNE_L1	SNE_VA1
Enzalutamide	40	40	40	40	40
SNEDDS Vehicles	1000	1000	1000	1000	1000
Eudragit E PO	-	40	-	-	-
HPMCP HP55	-	-	40	-	-
HPC-L	-	-	-	40	-
Kollidon VA64	-	-	-	-	40
Colloidal silica	500	500	500	500	500
(Acetone)	(20,000)	(20,000)	(20,000)	(20,000)	(20,000)
Total	1540	1580	1580	1580	1580

**Table 2 pharmaceutics-16-00457-t002:** Composition of ENZ S-SNEDDS formulation loaded with Kollidon VA64.

Formulation (mg)	SNE_VA1	SNE_VA2	SNE_VA3	SNE_VA4	SNE_VA5
Enzalutamide	40	40	40	40	40
SNEDDS Vehicles	1000	1000	1000	1000	1000
Kollidon VA64	40	20	80	160	240
Colloidal silica	500	500	500	500	500
(Acetone)	(20,000)	(20,000)	(20,000)	(20,000)	(20,000)
Total	1580	1560	1620	1700	1780

**Table 3 pharmaceutics-16-00457-t003:** PK parameters of ENZ powder and SNE_VA4 at an equivalent dose of 50 mg/kg in rats. Data are presented as the means ± standard deviation (*n* = 6).

PK Parameters	ENZ Powder	SNE_VA4
AUC_0–72_ (μg∙h/mL)	37.5 ± 6.4	274.4 ± 47.6
C_max_ (μg/mL)	1.4 ± 0.8	8.9 ± 2.0
T_max_ (h)	3.8 ± 1.7	4.3 ± 1.9
K_el_	0.02 ± 0.01	0.03 ± 0.02
t_1/2_	47.3 ± 25.7	26.7 ± 11.6

## Data Availability

Data are available on request due to restrictions, e.g., privacy or ethical restrictions.

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
