# Peer review of "The Impact of Polymers on Enzalutamide Solid Self-Nanoemulsifying Drug Delivery System and Improved Bioavailability"

_pharmaceutics, 2024, doi:10.3390/pharmaceutics16040457_

Round 1

Reviewer 1 Report

Comments and Suggestions for Authors

Dear authors,

In this study it was developed an enzalutamide - loaded solid self-nanoemulsifying drug delivery system containing Kollidon VA64 as a recrystalli-303 zation inhibitor, increasing the solubility of this prostate cancer medicine. This could improve, in the future, the therapeutic outcome of oncologic patients. Therefore I consider this study higly relevant and original. 

I have only few observations:

Abstract- please provide the explanation of the abbreviation SNEDDS at its first appearance in the text

24-31 the word administration repeats many times. Please try to reformulate

34 BCS please explain the abbreviation

101,102, 192, 263, 266 etc. please explain the abbreviations from the tables and figures in their legends

My best regards

Reviewer 2 Report

Comments and Suggestions for Authors

This is an interesting study about The Impact of Polymers on Enzalutamide Solid Self-Nanoemulsifying Drug Delivery System and Improved Bioavailability.  I suggest it for publication after the following points are well addressed.

1. Line 30-31, one review (Colloids and Surfaces B: Biointerfaces 222 (2023): 113043) should be included to support such a claim.

2. The information about the molecular weight and polydispersity of the polymers used in this study should be added.

3. The drug loading content and effienicy should be calculated and compared systematically. 

4. The quality of figue 8 should be improved to a higher level.

5. Formatting issues. Table 1, SiO2 to SiO2. Please check all.

Comments on the Quality of English Language

Minor editing of English language required

Reviewer 3 Report

Comments and Suggestions for Authors

Authors describe development of a new drug delivery system containing enzalutamide, based on a solidified nanoemulsion. Main effort of the authors has been focused on finding a solution to the problem of recrystallization of the drug during the spray-drying process. Recrystallization was successfully prevented by introducing polymer (copovidone) to the system. However, a few issues require explanation.

1.     There is no clear, what is the exact role of polymers: should they improve solubility of the drug in gastric or intestinal fluid? Or are they intended to prevent recrystallization of enzalutamide during the spray-drying process? Maybe both that functions are desirable?

2.     Drug release from dds is usually displayed in cumulative cumulative charts. Line in such graph can not decrease, and can not reach more than 100% of the drug stored in the carrier. How did You calculated percentages showed in figures 3, 4 and 5? Are they cumulative charts? I do not understand how such values could be obtained.

3.     Line 80 and 81: does D.W. mean distilled water? Why did You not use appropriate buffer solutions to obtain solutions with desired pH? How did You set pH of distilled water? Was it stable?

4.     How did You get crystalline and amorphous enzalutamide? Did You buy it or make on yourself?

5.     I suggest introduce a bit more significant information in the abstract. In my opinion it is to scanty.

6.     Tables 1 and 2 have the same description, but they refer to different stages of the study – therefore they should be more accurate.

7.     There is no information about statistical tests. What are the values in charts: arithmetic means, SD, SE? How many samples within group?

Comments on the Quality of English Language

Minor corrections needed

Round 2

Reviewer 3 Report

Comments and Suggestions for Authors

If I well get it, polymers in S-SNEDDS exerted two significant effects on the drug carriers performance:

1. ENZ remaining either molecularly dissolved or in an amorphous state in the S-SNEDDS directly after spray-drying process.

2. Recrystalization in aqueous environment (such as gastric fluid) was prevented and bioavailability was improved.

If my thinking is correct, I suggest more clearly mention that these two distinct effects polymers on drug performance take place. Preferably in the Introduction.

Comments on the Quality of English Language

Generally OK
